

# Comparison of machine learning and deep learning techniques in promoter prediction across diverse species

Nikita Bhandari[1], Satyajeet Khare[2], Rahee Walambe[3,4] and Ketan Kotecha[1,3]

[1] Computer Science, Symbiosis Institute of Technology, Symbiosis International (Deemed University), Pune, MH, India
[2] Symbiosis School of Biological Sciences, Symbiosis International (Deemed University), Pune, MH, India
[3] Symbiosis Centre for Applied Artificial Intelligence, Symbiosis International (Deemed University), Pune, Maharashtra, India
[4] Electronics and Telecommunication Dept, Symbiosis Institute of Technology, Pune, Maharashtra, India

Corresponding authors
Satyajeet Khare,
satyajeet.khare@ssbs.edu.in
Rahee Walambe,
rahee.walambe@scaai.siu.edu.in

## ABSTRACT

Gene promoters are the key DNA regulatory elements positioned around the transcription start sites and are responsible for regulating gene transcription process. Various alignment-based, signal-based and content-based approaches are reported for the prediction of promoters. However, since all promoter sequences do not show explicit features, the prediction performance of these techniques is poor. Therefore, many machine learning and deep learning models have been proposed for promoter prediction. In this work, we studied methods for vector encoding and promoter classification using genome sequences of three distinct higher eukaryotes viz. yeast (Saccharomyces cerevisiae), *A. thaliana* (plant) and human (Homo *sapiens*). We compared one-hot vector encoding method with frequency-based tokenization (FBT) for data pre-processing on 1-D Convolutional Neural Network (CNN) model. We found that FBT gives a shorter input dimension reducing the training time without affecting the sensitivity and specificity of classification. We employed the deep learning techniques, mainly CNN and recurrent neural network with Long Short Term Memory (LSTM) and random forest (RF) classifier for promoter classification at k-mer sizes of 2, 4 and 8. We found CNN to be superior in classification of promoters from non-promoter sequences (binary classification) as well as species-specific classification of promoter sequences (multiclass classification). In summary, the contribution of this work lies in the use of synthetic shuffled negative dataset and frequency-based tokenization for pre-processing. This study provides a comprehensive and generic framework for classification tasks in genomic applications and can be extended to various classification problems.

# INTRODUCTION

Accurate transcription of a gene requires RNA polymerase enzyme to recognize the start site of the gene and the end. One of the key regions involved in the transcriptional regulation

of RNA present at the start site is called promotor. A fundamental requirement for establishment of gene expression pattern and regulatory network is enabled by promoters. The selection of promoter is an important factor for genetic engineering. It is used to manipulate gene architecture and expression of genes under various conditions (*Singla-Pareek, Reddy & Sopory, 2001*). Promoter sequences have a gene-specific architecture which makes it hard to identify them computationally. A strong TATA box is present in a number of promoters of highly expressed genes. On the other hand, multiple groups of genes manifest into the TATA-less promoters. In the last decade, genomes of several organisms have been sequenced. Though the gene information has been computationally recognized, the size and functional features of the promoters are still left largely undetermined in newly sequenced genomes (*Umarov & Solovyev, 2017*).

Prediction of promoters can be achieved in various ways e.g., simple sequence alignment, content-based approach, or signal-based approach, etc. Matching gapped fingerprints of unlabelled sequences with labelled sequences is an example of a sequence alignment approach (*Gordon et al., 2003*; *Wang et al., 1999*). The signal-based approach considers promoter elements such as TATA box, CCAAT-box, etc. as signals for prediction and ignores the non-element portion of the sequence resulting in poor prediction performance (*Knudsen, 1999*). The content-based approach considers the frequency distribution of k-mer fragments, for example, considering region with a high frequency of CpG sites (*Ioshikhes & Zhang, 2000*; *Li, Chen & Wasserman, 2016*). CpGProD (*Ponger & Mouchiroud, 2002*), McPromoter (*Ohler, 2000*), CONPRO (*Liu & David, 2002*), Eponine (*Down & Tim, 2002*), FirstEF (*Davuluri, 2003*) are some examples of signal-based and content-based approaches used for eukaryotic promoter prediction. The tuning of sequence alignment completely relies on the systematic usage of reference sequences. This can degrade the performance of distantly related set of sequences (*Mathur, 2013*). Also, sequence alignment methods are relatively less effective due to their heuristic nature and high memory requirement for alignment of longer sequences (*Chowdhury & Garai, 2017*; *Gordon et al., 2003*). ConSite (*Sandelin, Wasserman & Lenhard, 2004*), rVISTA (*Loots et al., 2002*), PromH (*Solovyev & Shahmuradov, 2003*), FootPrinter (*Blanchette & Tompa, 2003*) are sequence alignment based promoter prediction applications.

Machine learning (ML) and deep learning (DL) techniques play an important role in this area as compared to regular statistical methods mentioned above. ML techniques have been used in a variety of applications of genomics such as identification of splice site (*Larrañaga et al., 2006*; *Nguyen et al., 2016*), promoters regions (*Anwar et al., 2008*; *Lai et al., 2019*; *Rahman et al., 2019*), classification of diseased related genes (*Díaz-Uriarte, 2008*; *Thi et al., 2008*), identification of transcription start site (TSS) (*Libbrecht & Noble, 2015*), identification of protein binding sites (*Pan & Yan, 2017*), recognition of genomic signals such as polyadenylation sites and translation initiation sites (*Kalkatawi et al., 2019*), disease diagnosis (*Manogaran et al., 2018*), transcriptomics analysis (*Karthik & Sudha, 2018*), drug discovery and repurposing (*Cheng et al., 2019*), identification of biomarkers (*Tabl et al., 2019*), etc. Though the applications of ML and DL techniques in the field of genomics are growing, accurate prediction of promoters is still one of the most challenging tasks in genomics (*Anwar et al., 2008*; *Oubounyt et al., 2019*; *Umarov & Solovyev, 2017*).

Various attempts of application of ML and DL algorithms have been made in promoter prediction. *E. coli* promoters provide a simple model system to study the promoter prediction. Specifically, $\sigma70$ promoters from *E. coli* were subjected to intense investigation in the following studies. In one of the studies, 669 promoter sequences, each with length of 80 nucleotides were analysed using a synthetic background and a feedforward neural network with three layers resulting in 96% precision (*Rani, Bhavani & Bapi, 2007*). In another study, an ensemble of Support Vector Machine, Linear Discriminant Analysis and Logistic regression was utilized which resulted in a classification accuracy of 86.32% for promoter sequences (*Rahman et al., 2019*). One of the studies has used 106 records of $\sigma70$ promoter and non-promoter sequences to train the model using CNN. For vectorization of input sequences, they applied one hot encoding with the k-mer size of 3 nucleotides. This study came up with an accuracy of 99% (*Nguyen et al., 2016*). However, the smaller size of the dataset for deep learning remains a concern. A promoter predictive model based on CNN(PPCNN) achieved much better sensitivity and specificity on *E. coli* $\sigma70$, Arabidopsis and human promoters (*Umarov & Solovyev, 2017*). However, the sample size of the promoter and non-promoter sequences was variable for these organisms (81 base pairs for *E. coli* $\sigma70$ promoter whereas 251 base pairs for other organisms). Another approach employed SVM to discriminate promoters and non-promoters of five different organisms (plants (various species), Drosophila, Homo *sapiens*, Mus *musculus*, Rattus *norvegicus*) using k-mer size 4 (*Anwar et al., 2008*). With the test set of 100 sequences (50 promoter and 50 non-promoter sequences) for each species, an average accuracy of 88%, sensitivity of 86% and specificity of 87.6% was achieved.

Considering the previous work in the area of sequence classification into promoter and non- promoter categories using ML and DL methods, there is scope for further improvements in terms of prediction performance. Factors such as sequence length, k-mer size, selection of negative dataset, feature encoding technique will help to achieve the accurate prediction of promoter sequences. Also, the highly imbalanced positive and negative sample dataset is one of the major problems in promoter recognition as it leads to model overfitting and makes the model less generic. Additionally, randomly selecting a non-promoter region from the same genome as negative datasets has its own limitation as the training model tends to find very simple features. Several studies have used one-hot encoding for the sequence encoding process (*Giosue & Di Gangi, 2017*; *Lai et al., 2019*; *Nguyen et al., 2016*; *Oubounyt et al., 2019*; *Rahman et al., 2019*).

The objective of this study is to overcome the limitations listed above and to make a classifier more robust and generic. We collected the promoter and non-promoter sequences of three distinct organisms: Yeast, *A. thaliana*, Human and developed an effective and powerful promoter classifier using deep convolutional neural network. We have also used the frequency-based tokenization approach instead of one-hot encoding for feature vectorization for various k-mer sizes (2-mer, 4-mer, 8-mer). The classifier successfully distinguished promoter from non-promoter sequences with very high sensitivity, specificity, and accuracy. The same CNN model was used for the cross-species evaluation and multi-species classification of promoter sequences. CNN showed high efficiency in promoter prediction when compared with the LSTM and RF classifier.

This technique can be extended automatically for the recognition of complex functional elements in sequence data from biological molecules.

## METHODS

### Selection of dataset

We selected yeast, *A. thaliana* and human genomes for our analysis. The genomes were selected using UCSC genome browser (*Haeussler et al., 2019*). We collected the datasets of approximately 41,671 sequences of *A. thaliana* (UCSC version araTha1), 61,546 sequences of humans (UCSC version hg38) each, and 6125 sequences of yeast (UCSC version sacCer3). However, after pre-processing and data cleaning, we randomly selected 35,000, 35,000 and 6000 sequences for A. thaliana, human and yeast respectively for experimentation. One thousand basepairs (bp) long putative promoter regions ($-700$ to $+300$ bp around TSS) were extracted in two steps as follows: As a first step, 700 bases upstream of the TSS were selected in the Table Browser using RefSeq Genes as an input to create a custom track. As a second step, 300 bases downstream were selected using custom track as an input to extract a 1,000 bases promoter sequence in a FASTA format. Equal number of background sequences were generated using a synthetic method of shuffling of promoter sequences for each organism (*Caballero et al., 2014*). The ratio of positive and negative sequences for each organism was 1:1. Each of these datasets were processed for the development of distinct models. The number of samples were divided into training and testing sets with the split of 90% and 10%, respectively. Out of 90% of training samples 10% of data was considered for validation and parameter tuning. We have utilised stratified random sampling for splitting of data, and therefore equal percentage of samples of each target class were used for training and testing of models. These datasets were further subjected to feature extraction, feature encoding and classification purpose (Fig. 1). Number of learning epochs used were 10 and the batch size was 128. Binary cross-entropy and sparce categorical cross-entropy were used as loss function for binary and multiclass classification, respectively. Adam optimizer was utilized with default learning rate. Early stopping callbacks was monitored through loss on the validation dataset. The efficiency of the models was also tested using human sequence data. 1,000 nucleotides long sequences were extracted from the human genome (hg38) using fasta-subsample script of MEME Suite (v 5.0.5) (*Bailey et al., 2009*). These sequences were used as negative dataset to test the efficiency of the model.

### Data pre-processing

DNA sequences are 1D-channel of four nucleotides, adenine (A), thymine (T), guanine (G) and cytosine (C). Feature extraction process is used to generate input for ML/DL models from raw data of nucleotide sequences. For feature extraction we used k-merization process. In this process, a short sequence segment of k consecutive nucleotides (k-mers) with the stride of 1 were generated from parent sequences (*Chor et al., 2009*). We specifically studied the effect of feature-extraction parameter k-mer size on the prediction performance of the models. We used four different k-mer sizes for experimentation: 1-mer, 2-mer, 4-mer and 8-mer. Further, k-merized sequences were given as input to the feature encoding techniques.

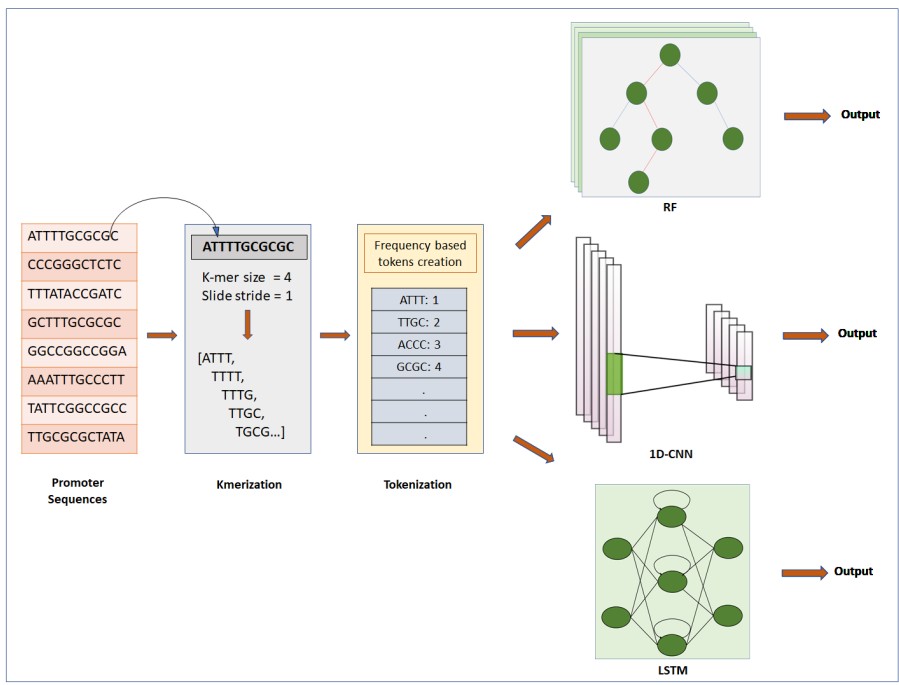

**Figure 1** **Steps for classification of DNA sequence: feature extraction, feature encoding and classification.** The first step is to input Raw sequences for feature extraction and feature encoding using K-merization and frequency-based tokenization techniques, respectively. Then the tokenized sequence fed to ML or DL model for further analysis. For deep learning, CNN and LSTM are used, whereas for machine learning RF is used.

## Feature encoding

We used two different approaches for feature encoding viz. frequency-based tokenization and one-hot encoding. In frequency-based tokenization, each k-mer in the sequence is encoded into tokens based on its frequency and mapped to a unique index starting from 1. Zero is a reserved index not assigned to any k-mer. This helps transform each sequence of k-mers to a sequence of integers. Feature encoding was achieved using in-house script written in Python (version 3) with system specification as: 64bit operating system, x64 based processor, 8GB RAM, intel CORE i7 processor. We used a text tokenization utility class of Keras pre-processing and data augmentation module for feature encoding. Whereas for one-hot encoding each k-mer is represented by a new binary variable. For this we utilized the OneHotEncoder class of Scikit-learn preprocessing utility. The output of Frequency based tokenization and one-hot encoding were subjected to CNN based analysis. The output of frequency-based tokenization was also subjected to other ML and DL analyses viz. random forest and LSTM RNN.

## Random forest classifier

Random forest is an ensemble of decision trees constructed using a different sample from the original data (*Breiman, 2001*). First, each tree is built from a random bootstrapped sample of the training data. Second, at each split of the tree, RF model considers only

a small subset of features for training. This helps to improve generalization by reducing variance. The final classification is obtained by combining results from the decision trees passed by votes. The bagging strategy of RF can effectively decrease the risk of overfitting when applied to large dimension data. Therefore, RF can handle a large data set with high dimensionality. Random forest has been widely used in the prediction of DNA-binding proteins (*Wang, Yang & Jack, 2009*), microarray data analysis (*Yang et al., 2010*), regulatory elements prediction (*Li et al., 2017*), etc. The encoded data are given as input to the Random Forest Classifier of scikit-learn ensemble module for classification of promoters and non-promoter sequences (*Pedregosa et al., 2011*). The value of n_estimator (number of trees in the forest) was 300, minimum number of samples required to split an internal node was 2 and bootstrap value was false.

## LSTM-recurrent neural network

For sequential data, the flow of gradients for long durations can help in learning long-term dependencies. This can be achieved using Long Short-Term Memory (LSTM) RNN (*Hochreiter & Schmidhuber, 1997*). We fed the pre-processed encoded input to the LSTM layer. The output of the LSTM layer with 128 units was fed to a Dense layer and the output of the Dense layer with 64 units was fed to the Drop-out layer and used Relu activation. In the Drop-out layer half the units were dropped. The final state was mapped through a fully connected layer with Sigmoid activation. We used LSTM, Activation, Dense, Dropout and Embedding methods from layer module of Keras (version: 2.2.4) to develop this architecture. Performance evaluation parameters are further used to evaluate the performance of the LSTM model.

## Convolutional neural network

Encoding helps to represent 1-D channel of DNA sequence in the form of a sequence of numerical values. This form of representation can be used as input to convolutional neural networks (*Collobert & Weston, 2008*; *LeCun et al., 1998*). To build CNN architecture, we utilized three 1D convolution layers, each followed by a max-pooling layer. Each convolutional layer uses filter of size 5. The pool size of the max-pooling layer was 4 with stride 1. The output of the max-pooling layer was then fed to three fully connected dense layers, consisting of 1,025, 512, 128 units and used the ReLu activation function with a 20% dropout. The final one is the classification layer and uses the sigmoid activation. We utilized Conv1D and MaxPooling1D methods from the layer module of Keras (version: 2.2.4) for experimentation. Performance evaluation parameters are further used to evaluate the performance of the CNN model.

## Performance evaluation parameters

After the application of the prediction model on the dataset, we obtained true positive (TP) and true negative (TN) numbers of truly identified promoter and non-promoter sequences. We also obtained false positive (FP) and false-negative (FN) numbers of falsely identified promoter and non-promoter sequences. In order to evaluate the performance of classification models we computed accuracy (Acc), sensitivity (Sn), specificity (Sp) (*Skaik,*

_2008_) and Matthews correlation coefficient (MCC) (_Matthews, 1975_) using following equations:

$$Accuracy = \frac{TP + TN}{TP + TN + FP + FN}$$

$$Sensitivity = \frac{TP}{TP + FN}$$

$$Specificity = \frac{TN}{TN + FP}$$

$$MCC = \frac{((TP * TN) - (FP * FN))}{\sqrt{((TP + FP) * (TP + FN) * (TN + FP) * (TN + FN))}}.$$

## RESULTS

### Selection of genomic datasets

We selected three eukaryotic genomes viz. yeast, _A. thaliana,_ and humans, for classification of sequences into the promoter and non-promoter groups. Yeast being the smallest genome, resulted in a small dataset of ~6,000 promoters. Large genomes of _A. thaliana_ and humans generated datasets of ~35,000 promoters each. Negative datasets were generated by reshuffling the promoter sequences. The data were divided into train and test sets with a split of 90% and 10%, respectively. The 10% of training data was used for validation and parameter tuning. The resulting datasets were used for comparison of feature encoding techniques and training ML/DL models. In order to evaluate the performance of the ML/DL models, a dataset of 600 random sequences from human genome was used.

### Comparison of feature encoding techniques

We tested efficiency and computational requirement of one-hot encoding and frequency-based tokenization (FBT) techniques. For this purpose, we used yeast dataset and tested the feature encoding methods on k-mer sizes 1 and 2 (Table 1). We found that at both k-mer sizes the training time was higher for one-hot encoding without significant improvement in sensitivity and specificity. For one-hot encoding, the testing accuracy of CNN model was 95% for 1-mer and 96% for 2-mer whereas for FBT accuracy achieved was 97% for 1-mer and 96% for 2-mer. We also trained higher k-mer sizes (more than 2-mer) however the training time required was very high at the described configuration. Based on these results and the fact that we wanted to use k-mer sizes 2, 4 and 8; we decided to use FBT for further work.

### Comparison of ML/DL algorithms in binary classification

The first set of experiments was conducted to perform binary classification of sequences into the promoter and non-promoter groups. The performance of CNN, LSTM and RF are shown in Table 2 for the yeast, _A. thaliana,_ and human datasets. The average performance

**Table 1  Input dimensions and training time for corresponding encoding techniques and k-mer sizes.**

| Techniques | Training time (min) | Input vector size | Acc | Sn | Sp | MCC |
|---|---|---|---|---|---|---|
| **1-mer** | | | | | | |
| One Hot Encoding | 28 | 4 X 1000 | 0.95 | 0.98 | 0.90 | 0.90 |
| Frequency Based Tokenization | 14 | 1 X 1000 | 0.97 | 0.98 | 0.99 | 0.97 |
| **2-mer** | | | | | | |
| One Hot Encoding | 240 | 16 X 999 | 0.96 | 0.97 | 0.95 | 0.93 |
| Frequency Based Tokenization | 14.3 | 1 x 999 | 0.96 | 0.98 | 0.93 | 0.89 |

**Table 2  Performance of CNN, LSTM and RF models for binary classification using different statistical measures.**

| Methods | Yeast | | | | *Arabidopsis thaliana* | | | | Human | | | |
|---|---|---|---|---|---|---|---|---|---|---|---|---|
| | Acc | Sn | Sp | MCC | Acc | Sn | Sp | MCC | Acc | Sn | Sp | MCC |
| **2-mer** | | | | | | | | | | | | |
| CNN | 0.96 | 0.98 | 0.93 | 0.89 | 0.98 | 0.97 | 0.99 | 0.95 | 0.99 | 0.99 | 0.99 | 0.99 |
| LSTM | 0.64 | 0.56 | 0.71 | 0.27 | 0.50 | 0.50 | 0.50 | 0.01 | 0.70 | 0.69 | 0.74 | 0.49 |
| RF | 0.87 | 0.84 | 0.90 | 0.74 | 0.73 | 0.76 | 0.70 | 0.46 | 0.72 | 0.76 | 0.68 | 0.45 |
| **4-mer** | | | | | | | | | | | | |
| CNN | 0.97 | 0.99 | 0.95 | 0.91 | 0.99 | 1.00 | 0.99 | 0.98 | 0.99 | 1.00 | 0.99 | 0.99 |
| LSTM | 0.86 | 0.91 | 0.82 | 0.73 | 0.88 | 0.88 | 0.90 | 0.78 | 0.93 | 0.94 | 0.95 | 0.89 |
| RF | 0.81 | 0.74 | 0.88 | 0.62 | 0.79 | 0.80 | 0.79 | 0.59 | 0.79 | 0.80 | 0.80 | 0.54 |
| **8-mer** | | | | | | | | | | | | |
| CNN | 0.95 | 0.95 | 0.96 | 0.91 | 0.99 | 0.99 | 0.99 | 0.98 | 0.98 | 0.99 | 0.98 | 0.98 |
| LSTM | 0.79 | 0.82 | 0.77 | 0.58 | 0.98 | 0.97 | 0.98 | 0.97 | 0.99 | 0.99 | 0.99 | 0.98 |
| RF | 0.73 | 0.66 | 0.81 | 0.47 | 0.85 | 0.82 | 0.88 | 0.69 | 0.84 | 0.81 | 0.87 | 0.69 |

of the CNN model on the test set of sequences was: Sn-95.33%, Sp-97.5%, Acc-96%, Mcc-92.5%. It was found that the computed CNN model performed better than the LSTM and RF for all k-mer sizes. For k-mer size 2, RF performed better than LSTM. However, LSTM tends to work significantly better for k-mer sizes 4 and 8 (Figs. 2A–2C). Then we analysed the efficiency of these models on each of the three organisms. In all three species, CNN outperformed LSTM and RF in all evaluation metrics (Figs. 2D–2F). Both CNN and RF do not show any pattern in change in the accuracy with k-mer size. Accuracy of CNN remains high throughout for all k-mer sizes. Accuracy of LSTM on the other hand improves with increase in k-mer size. However, we have observed that the computational requirement for LSTM increases with increase in k-mer size.

## Cross species evaluation using ML/DL models

We also performed a cross-species evaluation using CNN and RF. The purpose of the cross-species evaluation was to test whether sequence structures that underlie promoter regions are conserved across species (*Lai et al., 2019*). In cross-species evaluation, we trained the CNN and RF models using one organism's data and evaluated on the rest of the organisms for all k-mer sizes considered. We can see that the average accuracy performance

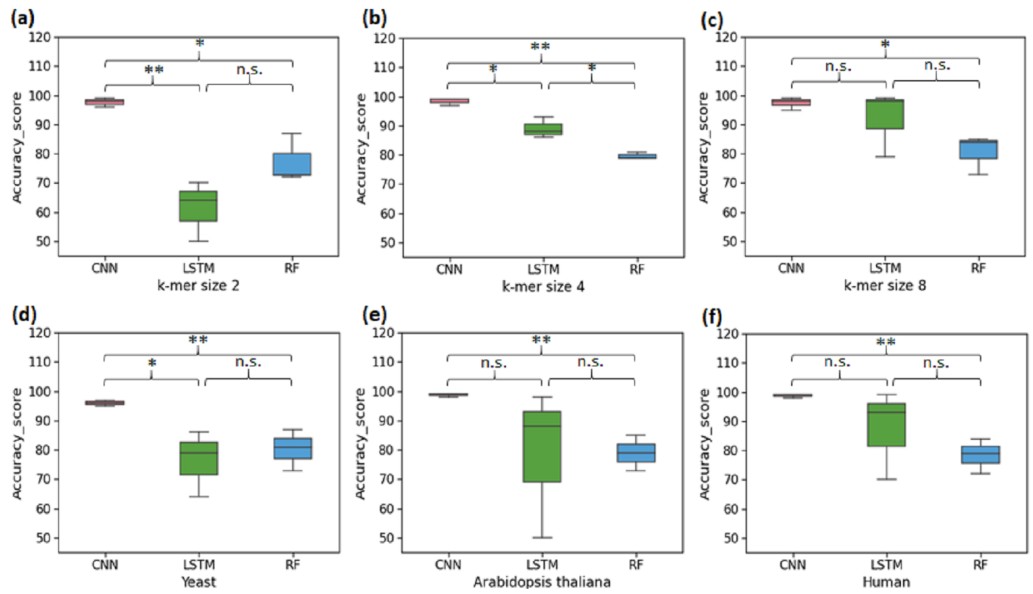

**Figure 2** **Performance analysis of CNN, LSTM, and RF for binary classification.** Three distinct organisms (yeast, *Arabidopsis thaliana*, human) were used. (A), (B) and (C): The accuracy performance of CNN, LSTM, and RF for various k-mer sizes. A significant difference in the performance of DL and ML techniques for each k-mer size was observed. (D), (E) and (F): The accuracy performance of CNN, LSTM, and RF on each species. A significant difference in the performance of DL and ML techniques for each organism was observed. (* statistically significant, ** statistically more significant, n.s. non-significant).

of CNN dropped to 72.77% and that of RF dropped to 55.66%. This result suggests that there is a large variation in sequence structures of promoters across species (Fig. 3).

## Testing of ML/DL model with random genome sequences

Finally, in order to evaluate the performance of these ML/DL models, we also used random sequences from human genome as negative data with testing data. The prediction performance of CNN model was better than LSTM and RF. The average accuracy achieved by CNN model is 91%, along with 88% sensitivity, 99% specificity and 85% Mcc (Table 3).

## Comparison of ML/DL algorithms in multiclass-classification

From cross-species evaluation we found that promoters from the different species have different sequence structures. Therefore, we performed the multispecies classification of sequences to a promoter category. We created a sample dataset of ~66,000 promoter sequences from yeast, *Arabidopsis thaliana,* and human. Multiclass classification performance of CNN, LSTM, and RF models for various k-mer sizes are shown in Table 4. The average prediction accuracy score of the CNN model is 98% whereas the average prediction accuracy score for LSTM is 96% and RF is 80%. CNN achieved the highest sensitivity and specificity among all techniques. The CNN and LSTM architectures outperformed the RF classifier for all k-mer sizes (CNN model: Sn-97%, Sp-99% MCC-97%; LSTM model: Sn-92%, Sp-97%, MCC-94%; RF: Sn-58%, Sp-87%, MCC-65%). For multiclass classification of sequences into promoter groups, the performance of RF

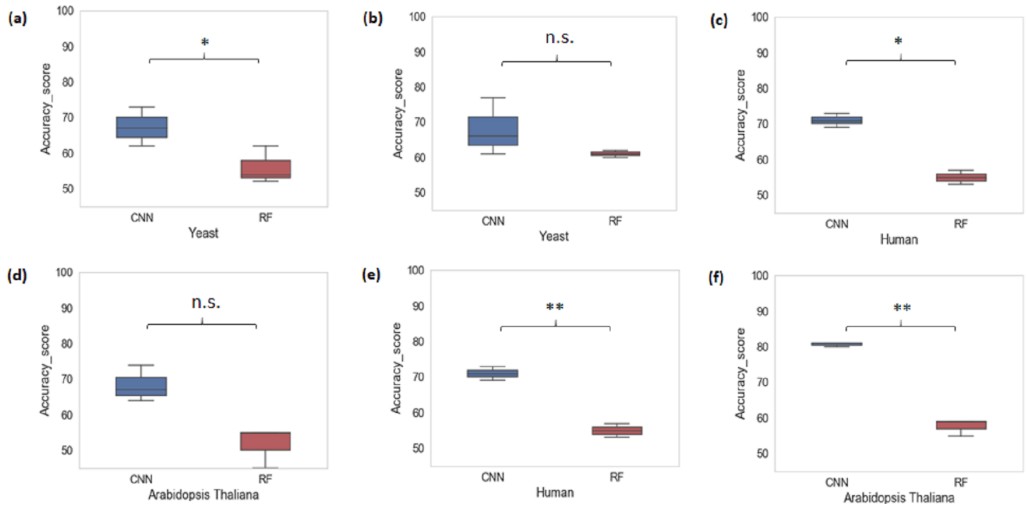

**Figure 3  Performance analysis of CNN and RF for cross-species evaluation.** (A & D) Trained on human data and tested on yeast and *Arabidopsis thaliana*. (B & E) Trained on *Arabidopsis thaliana* and tested on yeast and human. (C & F) Trained on yeast and tested on human and Arabidopsis thaliana. A significant difference in the performance of DL and ML techniques was observed.

**Table 3  Performance of CNN, LSTM and RF models for binary classification when random sequences from human genome are used as negative class data with test data.**

| Methods | Human | | | |
|---|---|---|---|---|
| | Acc | Se | Sp | MCC |
| | | 2-mer | | |
| CNN | 0.9 | 0.85 | 0.96 | 0.84 |
| LSTM | 0.56 | 0.09 | 0.96 | 0.01 |
| RF | 0.75 | 0.77 | 0.72 | 0.49 |
| | | 4-mer | | |
| CNN | 0.91 | 0.88 | 0.99 | 0.85 |
| LSTM | 0.89 | 0.82 | 0.96 | 0.79 |
| RF | 0.79 | 0.77 | 0.81 | 0.58 |
| | | 8-mer | | |
| CNN | 0.91 | 0.85 | 0.98 | 0.83 |
| LSTM | 0.91 | 0.84 | 0.99 | 0.84 |
| RF | 0.82 | 0.77 | 0.86 | 0.64 |

in terms of accuracy and sensitivity has decreased with an increase in k-mer size. LSTM achieved the highest sensitivity and specificity at k-mer 4. The deep learning techniques CNN and LSTM outperformed RF (Fig. 4A), whereas change in the k-mer size does not show a significant difference in the prediction performance (Fig. 4B).

**Table 4 Performance of CNN, LSTM and RF models for multi-species classification using different statistical measures.**

| Methods | Acc | Sn | Sp | MCC |
|---|---|---|---|---|
| **2-mer** | | | | |
| CNN | 0.98 | 0.97 | 0.99 | 0.97 |
| LSTM | 0.96 | 0.93 | 0.94 | 0.93 |
| RF | 0.84 | 0.61 | 0.89 | 0.73 |
| **4-mer** | | | | |
| CNN | 0.99 | 0.98 | 0.99 | 0.98 |
| LSTM | 0.98 | 0.96 | 0.99 | 0.96 |
| RF | 0.83 | 0.60 | 0.89 | 0.70 |
| **8-mer** | | | | |
| CNN | 0.98 | 0.96 | 0.99 | 0.97 |
| LSTM | 0.95 | 0.86 | 0.97 | 0.92 |
| RF | 0.74 | 0.54 | 0.83 | 0.52 |

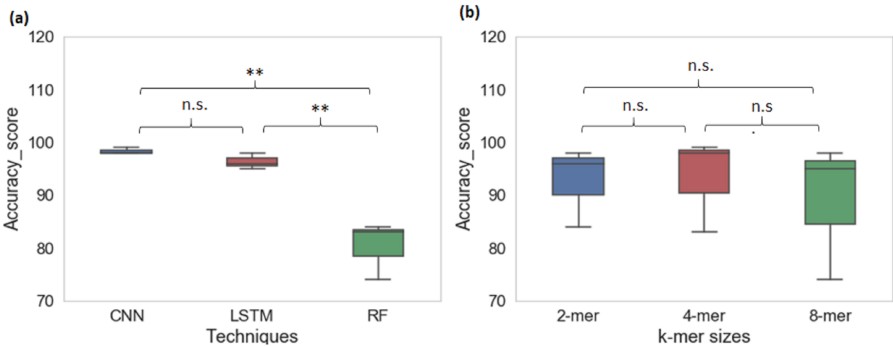

**Figure 4 Performance analysis of CNN, LSTM, and RF for multi-species promoter sequence classification.**

## DISCUSSION

A common practice in DNA sequence analysis is to use a set of 'background' sequences as negative controls for evaluation of the false-positive rates of gene recognition techniques for the detection of cis-regulatory elements. Generally, in the computational analysis of gene sequences, a set of non-target sequences extracted from the same genome is used as a background for evaluation purposes (*Rani, Bhavani & Bapi, 2007*). However, randomly selecting a non-promoter region from the same genome as a negative dataset may have its own limitations. Such a model can easily find basic features to separate two classes but not the less obvious ones. Another approach to create a background dataset is shuffling of the nucleotide sequences of a fraction of positive dataset sequences. In our analysis, we found that such a synthetic background dataset of shuffled sequences resulted in models with high sensitivity and specificity of classification.

We used frequency-based tokenization of k-mers instead of one-hot encoding (*Giosue & Di Gangi, 2017*; *Nguyen et al., 2016*; *Umarov & Solovyev, 2017*) for feature encoding. In

one-hot encoding each input sequence is thought as a matrix of 0's and 1's with the size $4^k \times L$, where k is the length of k-mer and L is the length of the sequence. This input to embedding layer is highly sparse. The input dimension increases with an increase in k-mer which will automatically results in high computational processing time. Also, the one-hot encoded matrix representation of a sequence cannot capture the significance of the number of times the subsequence or motifs is going to occur. The advantage of frequency-based tokenization over one-hot encoding is that it gives a shorter input dimension to the AI model and can save training time significantly as compared to the one-hot encoding. Therefore, one-hot encoding may produce comparable results in terms of accuracy, however computational configuration required for implementation of one hot encoding for higher k-mer sizes may go beyond the computational support available with most of the researchers.

Optimization of parameters is crucial for sequence predictor construction. During the classification of sequences into a promoter and non-promoter categories, an empirically identified tuneable parameter was k-mer size along with the configuration of network architecture. With the change in k-mer size, the resulting feature vectors size also changed, affecting predictive performance of the models. We studied the effect of 2-mer, 4-mer and 8-mer on the prediction performance of CNN, LSTM and RF on distinct organisms. As mentioned earlier, the performance of the LSTM model improved with an increase in k-mer size. We also tried k-mer sizes of 12 and 16 for each of the models. However, this increased the number of training parameters exponentially resulting in a "resource exhaustion problem" due to the consumption of large amounts of memory. Further, study on the combination of feature-extraction, feature encoding and the model architecture parameters is needed which can yield improvements in the prediction performance and help to reduce the execution time of the program.

We used four performance evaluation parameters Accuracy, Sensitivity (recall), Specificity and MCC. Accuracy is an average prediction performance on sample datasets. Still, accuracy alone cannot be an accurate measure to evaluate an ML model. Therefore, sensitivity and specificity are used for measuring the fraction of true positives and true negatives that are correctly predicted. With CNN and LSTM, deep learning models we have achieved both high sensitivity and specificity in all organisms.

The results show that the performance of CNN is better than LSTM and RF for all distinct organisms. The CNN models have reduced the number of false-positive and false-negative predictions and achieved high accuracy in both binary and multiclass classification. It demonstrates the ability of CNN to identify and extract abstract complex functional features with least pre-processing. In the case of cross-species evaluation, the performance of CNN is better than RF. However, the performance of both models is low, as promoters from the different species have different sequence structures, composition, and regulatory mechanisms. However, considering the differences in the type of data and data size, the differences observed in accuracy, sensitivity, specificity and MCC may not necessarily reflect on the model developed.

## CONCLUSIONS

The primary aim of this work is to efficiently discriminate sequences into a promoter and non-promoter sequences with a high true positive rate and true negative rate along with higher accuracy. We have proposed and demonstrated three specific improvements to the traditional methods for developing the generic and robust framework for classification tasks in the genomic domain. For our analysis, instead of using randomly selected non-promoter region we have utilized shuffled synthetic promoter sequences as a negative dataset to achieve necessary heterogeneity and robustness. A set of random sequences from human genome were used to test the efficiency of the models. For pre-processing of data, we have used k-mer based subsampling and frequency-based tokenization of sequences for feature extraction and vector representation respectively ensuring the reduction in training time. The deep learning techniques, namely, CNN and LSTM are employed for the classification of sequences into promoters and non-promoter categories which is important to interpret the underlying working of gene regulation. These methods are independent of the identification of any elements such as TATA-box, GC-box, CpG islands and sequence alignment methods for promoter prediction which are traditionally employed for this task. Using CNN, we achieved both high sensitivity and specificity while achieving higher accuracy on such a huge dataset. Results show a superiority of the CNN architecture over LSTM and RF in the binary and multispecies sequence classification. The effect of k-mer size on the model, both in terms of performance and training time is also extensively demonstrated. The proposed improvements and the CNN based approach are extremely generic and can be utilised to identify other elements of gene sequences and to meet the requirements of molecular biologists.

### Funding

This work has been supported by the Scheme for Promotion of Academic and Research Collaboration (SPARC) 2018-19, MHRD (project no. P104). Nikita Bhandari was supported by the Junior Research Fellowship Award 2018 by Symbiosis International Deemed University, India. The funders had no role in study design, data collection and analysis, decision to publish, or preparation of the manuscript.

### Grant Disclosures

The following grant information was disclosed by the authors:
Scheme for Promotion of Academic and Research Collaboration (SPARC) 2018-19, MHRD:  project no. P104.
Symbiosis International Deemed University, India.

### Competing Interests

The authors declare there are no competing interests.

## Author Contributions

- Nikita Bhandari conceived and designed the experiments, performed the experiments, analyzed the data, performed the computation work, prepared figures and/or tables, authored or reviewed drafts of the paper, and approved the final draft.
- Satyajeet Khare conceived and designed the experiments, performed the experiments, analyzed the data, prepared figures and/or tables, authored or reviewed drafts of the paper, and approved the final draft.
- Rahee Walambe conceived and designed the experiments, analyzed the data, authored or reviewed drafts of the paper, and approved the final draft.
- Ketan Kotecha analyzed the data, authored or reviewed drafts of the paper, and approved the final draft.

## Data Availability

The putative upstream regulatory regions used for this study, list of tools, code snippet for reshuffling of sequences, k-merization of sequences, CNN and LSTM architecture, and model plot are available at GitHub: https://github.com/nikitabhandari-dl/Dataset.

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
