# Peer review of "Comparison of machine learning and deep learning techniques in promoter prediction across diverse species"

_PeerJ Computer Science, doi:10.7717/peerj-cs.365_

## Round 0.1 · original submission · Major Revisions

Based on the concerns and comments of two of the reviewers regarding the validity of the results, I recommend a major revision of the manuscript. Authors need to address all comments and follow recommendations to share data/programs for reproducibility.

Reviewer 1 ·

Basic reporting

No Comment

Experimental design

No Comment

Validity of the findings

No Comment

Additional comments

The paper is well written and addresses an important research problem. One of the key research problem identified is, as quoted in the paper, " the highly imbalanced positive and negative sample dataset is one of the major problems in promoter recognition as it leads to model overfitting and makes the model less generic." Author should explain more clearly about how this challenge is address in the preparation of the dataset and in conducting the experiments. Apart from that the experiments are well conducted and documented. Performance evaluation parameters are well know, could have been explain briefly with citations.

·

Basic reporting

Bhandari et al., proposed and compared the results obtained from one-hot vector encoding and frequency based tokenization on CNN/RF/LSTM models for the promoter classification in three organisms. The authors claim the prediction exceeds previous methods. The writing of the manuscript and flow of ideas could be improved.
1. First paragraph in the Results section may be redundant as some of these sentences repeat what has already been described in Methods.
2. The following are some questions related to some of the manuscript sentences.

Line 84 - Mention why these methods are less effective.
Line 90 - Some may argue that ML are statistical methods (e.g., Bayes). Give an example of which statistical methods you are referring to.
Line 116 - Use scientific names for the organisms not previously defined.
Line 117 - Not clear. Is the number of promoter and non-promoter sequences is equal to 100 or is it 100 for each?
Line 131 - Use consistent names and terms. Yeast was previously defined as Fungus, A. thaliana as Plant, etc.

Experimental design

The following are my remarks/concerns about this work.

1. Can you please explain how simple features are produced when using samples from the same genome as background/negative sequences? (line 125)
2. Considering the amount of samples (35,000 and 6,000), I would recommend to use cross-validation to assess the performance of the models. Are the positive/negative samples ratio 1:1, if so, what would be the results of testing the entire genome for TSS where the number of negative samples is much larger than positive samples. With this test, the functionality of the method would be demonstrated.
3. It is not clear how many of these samples are extracted from each organism as well as the number of positive/negative. In Methods section, it is not clearly defined if one model is derived from each organism or are these sequences pooled?
4. Finally, if a data split of 90/10 for training and testing was used, how do you use the 1,000 bp long human sequences? how many of these sequences are you using? Again, in results (line 225) it is not clear how these sequences are used, i.e., to test the performance of which model?
5. Can authors please explain if and how the hyperparameters of each ML/DL model were tuned? Did authors tune these parameters using only the training data? did authors use validation set?
6. In line 288, authors mentioned that using a synthetic background dataset resulted in model with high sensitivity and specificity. Could this be because of the methodology for shuffling these promoters regions? did shuffling cause a random distribution of nucleotides? For this, an evaluation of the models with real genome sequences would be needed.

Validity of the findings

1. The validity of the finding can be assessed only after understanding the details about the parameter tuning of the models, data split (training, validation, test), number of samples for each organism, and ratio of samples.
2. Table 5 shows the comparison of the proposed method against other related works. I would like to see the author following exactly what the comparing methods did (same training/validation/test data) to establish the advantage of the proposed method. I find it very difficult to compare models trained and tested on different datasets.
3. Data should be made available online for reproducibility purposes.

Additional comments

Writing and style.
The following are few recommendations to improve the writing:
Line 48 - Replace "We employed the deep..." by "We employed different deep..."
Line 48 - Replace "... mainly CNN and recurrent..." by "... mainly CNN, recurrent..."
Line 54 - Replace "for pre-processing." by "for data pre-preprocessing."
Line 64 - Do not capitalize words in the sentence. Replace "...Genetic..." by "...genetic..."
Line 64 - Avoid the use of dangling modifiers. Replace ..."engineering. It ..." by "...engineering as it ..."
Line 94 - Make sure abbreviations are defined before using them (e.g., TSS).
Line 102 - Rewrite sentence. Example: "..., 669 promoters of 80 nucleotides long..."
Line 110 - use abbreviations already defined. Replace "deep learning" by "DL".
Line 122 - add "and" before feature encoding.
Line 138 - replace "automatically" to "seamlessly"
Line 143 - bp not defined.
Line 183 - LSTM already defined. RNN not defined.
Line 198-199 - avoid dangling modifiers.

Punctuation marks
Review the following remarks in the entire manuscript.
Line 66 - Use "," before "which".
Line 73 - Use comma before and after e.g.
Line 74 and 77 - Consider replacing etc by ", among others."
Line 76 - Use comma before "such as"
Line 105 - Avoid capitalization of words in the middle of the sentences.
Line 147 - Add comma before respectively.
Use comma before the use of viz.

·

Basic reporting

• The report is well-written in a clear language and a good structure and flow of information
• In the introduction, the paragraph that starts at line 89, it is mentioning some of the genomics applications and not covering all of them; such information should be clearly state

Experimental design

• Data extraction was not explained well and needed more details
• 90/10 data division for training and testing is subject to overfitting, and the performance of the testing, in this case, is questionable
• Random Forest (RF) is not a representation of ML models, so elaborate on why you selected it?
• Ceasing your experiments, as mentioned in section Comparison of Feature Encoding Techniques at line 226, because the training time was very high is not enough to draw your conclusion
• Regarding comparing with existing techniques, it is not a fair comparison in which you should unify the datasets and no just report the published results

Validity of the findings

• You have mentioned one of the challenges that were in previous studies is choosing the negative data, yet I do not think shuffling it would solve the
• I think having 1000 long sequence is too long to the extent that you might lose some of the important features and include irrelevant ones

Additional comments

• You have mentioned one of the challenges that were in previous studies is choosing the negative data, yet I don’t think shuffling it would solve the
• Have you checked the sequences and removed the duplicates or 100% identical sequences?
• I would recommend a visual representation of your ML/DL models with the parameters that were used
• The subtitle “Feature extraction” might be changed to data representation or data preprocessing

---

## Round 0.2 · Major Revisions

One of the reviewers still has concerns about the validity of the assessment which is crucial for PeerJ Computer Science. I recommend the authors to provide all necessary details, and perform the required experiments if any. I do not believe the comments are sophisticated to address and therefore, I suggest you address them in a timely manner.

·

Basic reporting

Comments addressed by the authors.

Experimental design

Reviewer’s original comment 1:
Considering the amount of samples (35,000 and 6,000), I would recommend using
cross-validation to assess the performance of the models. Are the positive/negative
samples ratio 1:1, if so, what would be the results of testing the entire genome for
TSS where the number of negative samples is much larger than positive samples.
With this test, the functionality of the method would be demonstrated.

Authors’ response: We agree with the reviewer that the number of samples would require
performance assessment of the models for each species. Number of positive and
negative samples are indeed in ratio 1:1. In order to validate the performance
assessment we also tested the model generated using shuffled sequences as negative
dataset on a large number of randomly selected non-TSS sequences from human
genome. We found that the model functioned equally well with this dataset validating
efficacy of models (Please refer to the line number: 281).

Reviewer:
Can authors please explain why only 593 random sequences from the human genome were considered as negative test set?
Also, please be accurate in the dataset sizes. In the manuscript, it is mentioned approximately 35,000 promoter sequences for each A. Thaliana and Human, and 6,000 for yeast. However, in the files I found 41671, 61546, and 6125 for A. Thaliana, Human, and yeast, respectively.
Finally, for reproducibility of data extraction, please indicate genome references, annotation used, and any other considerations.
* * *
Reviewer’s comment 2:
Can authors please explain if and how the hyperparameters of each ML/DL model
were tuned? Did authors tune these parameters using only the training data? did
authors use validation set?

Authors’ response: Typically, hyperparameters are tuned using grid search method, but due to
the amount of data and challenges with preprocessing, we did hyperparameter tuning
empirically. Commonly tuned hyperparameters were: Network layers, Filter size,
Kernel regularization, Activation function, Loss function, Optimizer, Number of
epochs, Learning rate, Decay rate, Batch Size. The line number 281 of the revised
manuscript refers the cross-validation set.

Reviewer:
To be certain that no biases were introduced in the results (due to data leakage while tuning the models), it is essential to describe how the parameters of the models were tuned. Normally, a portion of the training data is exclusively reserved for tuning these parameters (validation set). Finally, the test portion of the data is only used once for reporting the final results of the model. If parameters were empirically set by using the test set directly, then there is a data leakage in the approach. Please clarify these details in the manuscript and add a tale with the final model parameters used.
* * *
Reviewer’s comment 3:
In line 288, authors mentioned that using a synthetic background dataset resulted
in model with high sensitivity and specificity. Could this be because of the
methodology for shuffling these promoters regions? did shuffling cause a random
distribution of nucleotides? For this, an evaluation of the models with real genome
sequences would be needed.

Author's response: The shuffling did cause a random distribution of nucleotides keeping the overall nucleotide composition same. As suggested by reviewer we have evaluated the
model with real genome sequence data as described in section “training of ML/DL
model with random genome sequences”. Please refer to the line number 281.

Reviewer:
In line 283, the authors mentioned that a ~66,000 sequences dataset combining the different considered organisms was created. If I understand correctly, the results discussed in lines 285-294 are obtained by using the same 90%/10% data split for training/testing. If so, and related to my previous (parameter tuning), if the parameters were empirically set by optimizing the performance on the test set (without a validation set) then results would be biased and models would not necessarily generalize to unseen sequences with the reported results.

Validity of the findings

Reviewer’s original comment 4:
Considering the number of samples (35,000 and 6,000), I would recommend using
cross-validation to assess the performance of the models. Are the positive/negative
samples ratio 1:1, if so, what would be the results of testing the entire genome for
TSS where the number of negative samples is much larger than positive samples.
With this test, the functionality of the method would be demonstrated.

Authors' response: We agree with the reviewer that the number of samples would require
performance assessment of the models for each species. Number of positive and
negative samples are indeed in ratio 1:1. In order to validate the performance
assessment we also tested the model generated using shuffled sequences as negative
dataset on a large number of randomly selected non-TSS sequences from human
genome. We found that the model functioned equally well with this dataset validating
efficacy of models (Please refer to the line number: 281).

Reviewer:
I still consider 10% of testing data from an already small dataset may not represent the generalization capabilities of the model. Moreover, 593 random human non-promoter sequences may be too few. This is especially important to validate considering that no details are given on how parameters were tuned during the training of the model and which data was used.

Additional comments

The authors addressed most of my comments and I am overall satisfied with the results. However, I have concerns that are critical to be addressed. Below, I write my original comment, followed by Author’s response and my current concerns.

·

Basic reporting

The comments were handled.

Experimental design

The provided answers were good enough.

Validity of the findings

The issues were clarified.

---

## Round 0.3 · Minor Revisions

One of the reviewers is still having concerns regarding the level of provided details. Please address the comment and consider applying as soon as possible.

·

Basic reporting

The authors addressed all of my concerns.

Experimental design

Please note that methods should be described with sufficient information to be reproducible by other researchers. Having said that, data selection and pre-processing remain unclear in the manuscript.
From the last reply from the authors:
"The dataset uploaded were the 41671, 61546, and 6125 in size. However, after preprocessing and data cleaning process the dataset were reduced to 35949, 37957 and 6042, respectively. Out of these, for experimentation we have randomly selected 35000, 35000 and 6000 sequences for A. thaliana, human and yeast, respectively."

I still consider these details need to be clarified in the manuscript.
Moreover, the data available at GitHub only contain the raw sequences without any preprocessing. I suggest authors explain this process in the manuscript and also upload a "raw data" and "processed data" folders in the GitHub link.

Validity of the findings

The authors addressed all of my concerns.

Additional comments

I am satisfied with the reviewed manuscript.

---

## Round 0.4 · accepted · Accept

The authors have addressed the comments of all reviewers. I recommend the paper be published in PeerJ Computer Science.